# SGLT2 Inhibitors in COVID-19: Umbrella Review, Meta-Analysis, and Bayesian Sensitivity Assessment

**DOI:** 10.3390/diseases13030067

**Published:** 2025-02-21

**Authors:** Vinay Suresh, Muhammad Aaqib Shamim, Victor Ghosh, Tirth Dave, Malavika Jayan, Amogh Verma, Vivek Sanker, Priyanka Roy, Mainak Bardhan

**Affiliations:** 1King George’s Medical University, Lucknow 226003, India; 2Department of Pharmacology, All India Institute of Medical Sciences, Jodhpur 342005, India; 3Andhra Medical College, Visakhapatnam 530002, India; 4Bukovinian State Medical University, 58002 Chernivtsi, Ukraine; 5Department of Internal Medicine, Bangalore Medical College and Research Institute, Bangalore 560002, India; 6Department of Internal Medicine, Rama Medical College Hospital and Research Centre, Hapur 245304, India; 7Department of Neurosurgery, Trivandrum Medical College Hospital, Trivandrum 695011, India; 8Department of Labour, Government of West Bengal, Kolkata 700001, India; 9The Dr. John T. Macdonald Foundation, Department of Human Genetics, University of Miami Miller School of Medicine, Miami, FL 33136, USA

**Keywords:** SGLT2 inhibitors, SGLT2i, COVID-19, type 2 diabetes mellitus, meta-analysis, Bayesian sensitivity, mortality, hospitalisation

## Abstract

Background: Several studies have reported a reduced risk of COVID-19-related mortality in patients taking antidiabetic medications. This is an umbrella review, meta-analysis, and Bayesian sensitivity assessment of SGLT2 inhibitors (SGLT2is) in COVID-19 patients with type 2 diabetes mellitus (T2DM). Methods: A search was conducted on the MEDLINE (PubMed), EMBASE, Cochrane, and ClinicalTrials.gov databases on 5/12/2023. We performed an umbrella review of systematic reviews and meta-analyses on the effects of SGLT2is in T2DM patients with COVID-19 and critically appraised them using AMSTAR 2.0. Trials investigating SGLT2i use in COVID-19 patients post-hospitalisation and observational studies on prior SGLT2i use among COVID-19 patients were included in the meta-analysis, adhering to the PRISMA guidelines. Results: SGLT2is exhibited significantly lower odds of mortality (OR 0.67, 95% CI 0.53–0.84) and hospitalisation (OR 0.84, 0.75–0.94) in COVID-19 patients with T2DM. Bayesian sensitivity analyses corroborated most of the findings, with differences observed in hospitalisation and mortality outcomes. SGLT-2 inhibitors showed an OR of 1.20 (95% CI 0.64–2.27) for diabetic ketoacidosis. Publication bias was observed for hospitalisation, but not for mortality. The GRADE assessment indicated a low to very low quality of evidence because of the observational studies included. Conclusions: The prophylactic use of SGLT2is reduces mortality and hospitalisation among COVID-19 patients, particularly in patients with diabetes. The utility of SGLT2is after hospitalisation is uncertain and warrants further investigation. A limited efficacy has been observed under critical conditions. Individualised assessment is crucial before integration into COVID-19 management.

## 1. Introduction

COVID-19, the fast-spreading pandemic caused by SARS-CoV-2, had infected more than 772 million people globally by the end of November 2023, killing approximately 6.9 million [1]. Most of these deaths were attributed to comorbid conditions such as hypertension, diabetes, cardiovascular disease, and respiratory disorders [2]. It has also been seen that patients with COVID-19 have a high prevalence of diabetes [3]. The risk of severe infection in patients with diabetes and COVID-19 is associated with an increase in angiotensin-converting enzyme 2 (ACE 2), impaired T Cell function, and increased interleukin-6 (IL-6), which promotes COVID-19 infection due to increased viral entry and an impaired immune response [3]. Patients with type 2 diabetes mellitus (T2DM) are more prone to hospitalisation, respiratory dysfunction, and COVID-19-related death [4,5,6]. Studies have shown that poor glycaemic control is significantly associated with a poor clinical prognosis in patients with T2DM and COVID-19 [7,8]. Therefore, there is a keen interest in exploring antidiabetic therapy to improve the clinical outcomes of individuals with T2DM who are affected by COVID-19 [7].

Antidiabetic agents, such as Metformin, Dipeptidyl Peptidase-4 (DPP-4) inhibitors, Sulfonylurea, Glinides, Sodium–Glucose Co-Transporter 2 inhibitors (SGLT2is), glucagon-like peptide-1 (GLP-1) Receptor Agonists, α-glycosidase inhibitors, and thiazolidinediones (TZDs), are effective, safe, and widely used in the treatment of diabetes. Apart from their primary function in managing diabetes, there have been documented reports of anti-inflammatory and antiviral properties associated with antidiabetic agents [9]. Previous studies have shown that antidiabetic medications may significantly reduce COVID-19-related mortalities. At the same time, insulin treatment is associated with poor outcomes [10,11,12,13]. SGLT2is are glucose-lowering agents used in T2DM with other comorbidities, such as chronic kidney disease (CKD) and heart failure [14,15]. However, evidence regarding the benefits of SGLT2is in T2DM patients with COVID-19 is limited. This study aimed to assess the effectiveness and safety of SGLT2is in reducing mortality and morbidity in patients with COVID-19.

## 2. Methodology

### 2.1. Literature Searching

MEDLINE (PubMed), EMBASE, Cochrane, and the National Library of Medicine (ClinicalTrials.gov) were searched. The PICO elements for the three research questions are listed in Appendix A.

There were three primary objectives during the literature search.
To identify studies investigating the effect of SGLT2is on COVID-19 patients and their outcomes after hospitalisation.To identify observational studies that have studied the effects of SGLT2is in patients who were already on the medication and later developed COVID-19.To identify systematic reviews and meta-analyses that have reviewed studies investigating the effects of SGLT-2 inhibitors and their outcomes in patients with diabetes diagnosed with COVID-19.

A search query was made for each database using a common framework. A protocol was pre-registered in PROSPERO (Registration: CRD42023493347).

### 2.2. Search Strategy

Search terms and their iterations were identified through Medical Subject Headings (MeSH) index terms and a text mining tool, PubMed PubReMiner [16]. Filters for observational study designs were adopted from the Harvard Countway Library guide [17], which is a modified version of the “fixed method B” proposed by Furlan et al. [18]. Filters for randomised trials in MEDLINE and EMBASE (sensitivity- and precision-maximising version) were adopted from the technical supplement to Chapter 4 of the Cochrane Handbook for Systematic Reviews of Interventions (version 6.2) [19]. A high-sensitivity and precision PubMed search filter developed by Salvador-Oliván et al. was used to identify systematic reviews and meta-analyses for an umbrella review [20]. This review adhered to the Preferred Reporting Items for Systematic Reviews and Meta-Analyses (PRISMA) guidelines (Appendix A). The search queries are listed in Appendix A. The search date ranged from the earliest available date to 7 December 2023. The inclusion and exclusion criteria are presented in Appendix A. 

### 2.3. Statistical Analysis

Relevant data were extracted from the studies as count data (numerator and denominator) or effect size (risk ratio or odds ratio with variability). Individual study estimates were pooled using the Mantel–Haenszel method [21] with a random-effects model, expecting clinical heterogeneity in the included studies. Heterogeneity was assessed using prediction intervals [22], *I*^2^ [23], and tau^2^ [24]. Subgroup analyses were performed based on the type of comparator used. Sensitivity analyses included a Bayesian meta-analysis with weakly informative priors (as these may provide better estimates when the number of studies is less [25]), leave-one-out meta-analysis, and subgrouping based on study quality. All analyses were performed using R version 4.3.0 [26] with a restricted maximum−likelihood estimator and Q−profile for tau^2^ and its CI.

The primary outcomes were as follows:

The odds ratio of mortality and hospitalisation among COVID-19 patients with a history of SGLT2i use compared to COVID-19 patients without a history of SGLT2i use.

### 2.4. Quality and Risk of Bias Assessment

The Revised Cochrane risk-of-bias tool for randomised trials (RoB 2) was used to assess the risk of bias analysis of randomised trials [27]. The Risk of Bias in Non-randomised Studies of Interventions (ROBINS-I) tool was used for observational studies [28]. The AMSTAR 2 tool was used to critically appraise the systematic reviews included in the umbrella review [29]. The class of evidence was rated based on the following criteria [30]:Convincing (class I): number of cases > 1000, *p* < 10^−6^, *I*^2^ < 50%, 95% prediction interval excluding the null, no small-study effects, and no excess significance bias;Highly suggestive (class II): when the number of cases was >1000, *p* < 10^−6^, the largest study with a statistically significant effect and class I criteria not met;Suggestive (class III): when the number of cases > 1000, *p* < 10^−3^, and class I–II criteria were not met;Weak (class IV) when *p* < 0.05,  and class I–III criteria were not met.Non-significant when *p* > 0.05.

## 3. Results

A total of 472 articles identified in the literature were screened after deduplication. Upon screening the titles and abstracts, 420 articles were deemed irrelevant. Subsequently, 52 articles underwent full-text assessment, with 32 exclusions for the following reasons: wrong study design (*n* = 12), wrong population (*n* = 7), wrong intervention (*n* = 3), irrelevant (*n* = 4), duplicate data (*n* = 4), unavailable results (*n* = 1), and wrong outcome (*n* = 1). Nine additional articles were incorporated through citation searching. The final review included 29 studies, comprising two randomised controlled trials, 17 observational studies, and 10 systematic reviews/meta-analyses. The screening process is illustrated in the PRISMA flowchart (Figure 1).

### 3.1. Population Characteristics

In the included studies, the majority [31,32,33,34,35,36,37,38,39,40,41,42,43,44,45,46,47,48,49] of participants belonged to the elderly age group, with mean ages ranging from 58 [33,34] to 86 years [42]. Most of the studies had an equal distribution of participants among the sexes [33,35,37,40,41,42,43,48], while some studies had a male-predominant population [31,32,34,38,39,49]. The characteristics of the population in the included studies are presented in Table 1. The umbrella review includes ten systematic reviews and meta-analyses; their characteristics are described in Table 2.

#### 3.1.1. Effect of SGLT2i Intervention on Hospitalised COVID-19 Patients (Trials)

The analysis of death from any cause revealed an estimated pooled risk ratio of 0.92 (CI: 0.76, 1.10, *I*^2^ = 18%) using the random-effects model (Figure 2a). The Bayesian sensitivity analysis of the trials yielded insignificant results (Figure 2b). The analysis of hospital discharge in one month revealed an estimated pooled risk ratio of 1.02 (CI: 1.00; 1.05, *I*^2^ = 0%) using the random-effects model (Figure 2c). The Bayesian sensitivity analysis of the trials yielded insignificant results (Figure 2d).

Analysis of mechanical ventilation within one month revealed an estimated pooled risk ratio of 1.17 (CI: 0.62; 2.24, *I*^2^ = 93%) using the random-effects model (Figure 3a). The analysis of renal replacement therapy within one month revealed an estimated pooled risk ratio of 0.84 (CI: 0.49; 1.44, *I*^2^ = 50%) using the random-effects model (Figure 3b). The analysis of acute kidney injury revealed an estimated pooled risk ratio of 0.93 (CI: 0.67, 1.28, *I*^2^ = 39%) using the random-effects model (Figure 3c). The analysis of ketoacidosis revealed an estimated pooled risk ratio of 2.97 (CI: 0.70; 12.58, *I*^2^ = 0%) using the random-effects model (Figure 3d).

#### 3.1.2. Prior SGLT2i Use Among COVID-19 Patients in Observational Studies

The analysis of mortality revealed an estimated pooled odds ratio of 0.67 (CI: 0.53; 0.84, *I*^2^ = 76%) using the random-effects model (Figure 4a). Subgroup analysis was conducted based on the comparators used in the studies. In comparisons between SGLT-2 inhibitors and GLP-1 receptor agonists/Dipeptidyl Peptidase 4 inhibitors concerning mortality, the pooled odds ratio was 0.58 (CI: 0.44; 0.77, *I*^2^ = 71%) (Figure 4b). When compared with non-SGLT2is overall, the pooled odds ratio was 0.87 (CI: 0.73; 1.02, *I*^2^ = 0%), and in comparison with non-dapaglifozin, it showed a pooled odds ratio of 0.25 (CI: 0.53; 0.84) (Figure 4b). The sensitivity analysis showed that no single study significantly affected the overall primary outcome (mortality). Removing any individual studies did not affect the significance of the results (Figure 4c). The Bayesian sensitivity analysis, as depicted in Figure 4d, indicated that, upon excluding specific studies, notable changes occurred in the outcomes.

The analysis of hospitalisation revealed an estimated pooled odds ratio of 0.84 (CI: 0.75; 0.94, *I*^2^ = 67%) utilising the random-effects model (Figure 5a). A subgroup analysis was conducted based on the comparators used in the studies. In comparisons between SGLT-2 inhibitors and GLP-1 receptor agonists/Dipeptidyl Peptidase 4 inhibitors concerning hospitalisation, the pooled odds ratio was 0.83 (CI: 0.72; 0.96, *I*^2^ = 72%) (Figure 5b). When compared with non-SGLT2is overall, the pooled odds ratio was 0.83 (CI: 0.67; 1.04, *I*^2^ = 69%). No single study significantly affected the sensitivity analysis’s overall outcome (hospitalisation) (Figure 5c). No individual study affected the significance of the results. Bayesian sensitivity analysis, however, revealed that the exclusion of certain studies led to significant shifts in the outcomes (Figure 5d).

Only Kahkoska et al. reported the outcomes of emergency visits, revealing an odds ratio of 0.96 (CI: 0.88;1.04) (Figure 6a). The analysis of ICU admission revealed an estimated pooled odds ratio of 0.93 (CI: 0.82; 1.06, *I*^2^ = 0%) using the random-effects model (Figure 6b). A subgroup analysis was conducted based on the comparators used in the studies. In comparisons between SGLT-2 inhibitors and GLP-1 receptor agonists/Dipeptidyl Peptidase 4 inhibitors concerning ICU admission, the odds ratio reported by Israelsen et al. was 0.99 (CI: 0.44; 2.26) (Figure 6c). When compared with non-SGLT2is overall, the estimated pooled odds ratio was 0.93 (CI: 0.82; 1.06, *I*^2^ = 0%) (Figure 6c).

The analysis of mechanical ventilation revealed an estimated pooled odds ratio of 0.82 (CI: 0.50; 1.36, *I*^2^ = 7%) utilising the random-effects model (Figure 6d). A subgroup analysis was conducted based on the comparators used in the studies. In comparisons between SGLT-2 inhibitors and GLP-1 receptor agonists/Dipeptidyl Peptidase 4 inhibitors concerning mechanical ventilation, the estimated pooled odds ratio was 0.83 (CI: 0.47; 1.46, *I*^2^ = 46%) (Figure 6e). When compared with non-SGLT2is, Dalan et al. reported an odds ratio of 0.54 (CI: 0.07; 4.13) (Figure 6e).

The analysis of diabetic ketoacidosis revealed an estimated pooled odds ratio of 1.20 (CI: 0.64; 2.27, *I*^2^ = 0%) using the random-effects model (Figure 7a). Min et al. reported odds ratios of 0.81 (CI: 0.44 to 1.49) for acute kidney injury and 1.71 (CI: 0.57 to 5.17) for hypoglycaemia as compared to GLP-1 receptor agonists/Dipeptidyl Peptidase 4 inhibitors (Figure 7b,c).

### 3.2. Risk of Bias Analysis

Risk of bias analysis for the trials was performed using ROB 2.0, with both trials being categorised as having “low bias” (Appendix A). The risk of bias in the observational studies was assessed using the ROBINS-I tool, as depicted in Appendix A. The risk of bias in the systematic reviews, assessed using the JBI tool, is shown in Appendix A.

### 3.3. Publication Bias

Publication bias was assessed for the outcomes “hospitalisation” and “mortality,” as other outcomes lacked sufficient studies for evaluation. The Doi plot displayed visual asymmetry (Appendix A), and the LFK index was −3.88, suggesting publication bias in studies reporting hospitalisation. However, the trim-and-fill contour-enhanced funnel plot with Egger’s regression revealed a *p*-value of 0.55, indicating no evidence of publication bias or small-study effects for studies that reported the outcome “mortality”.

### 3.4. GRADE Assessment

The GRADE assessment table outlines the evaluation of SGLT2 inhibitor use in COVID-19 (Appendix A). The certainty of evidence varied from low (hospitalisation: odds ratio 0.84, CI 0.75–0.94; mortality: odds ratio 0.67, CI 0.53–0.84) to very low (ICU admission: odds ratio 0.93, CI 0.82–1.06; mechanical ventilation: odds ratio 0.82, CI 0.50–1.36; emergency visits: odds ratio 0.96, CI 0.88–1.04; diabetic ketoacidosis: odds ratio 1.20, CI 0.64–2.27; and acute kidney injury: odds ratio 0.81, CI 0.44–1.49). Despite the critical importance of these outcomes in COVID-19 management, the uncertainty of the data persists due to limitations in the study design, risk of bias, inconsistency, and imprecision observed in non-randomised studies.

## 4. Discussion

Our meta-analysis aimed to comprehensively review the utility of SGLT-2 inhibitors in COVID-19 patients. Focusing on their role as an intervention for hospitalised COVID-19 patients, our analysis centred on the following two randomised controlled trials: the RECOVERY and DARE-19 trials [31,32]. The analysis revealed no significant differences between the two groups in terms of mortality, hospital discharge, the need for mechanical ventilation, renal replacement therapy, acute kidney injury, or ketoacidosis associated with SGLT2 inhibitor use. Both the RECOVERY and DARE-19 trials reached similar conclusions, although their inclusion criteria varied [31,32]. The RECOVERY trial did not include patients who had diabetes at the time of recruitment, but did include patients with a history of diabetes [31]. In contrast, the DARE-19 trial included participants exhibiting cardiometabolic risk factors, including type 2 diabetes, heart failure, and chronic kidney disease [32]. The presence of comorbidities, particularly diabetes, heart failure, and chronic kidney disease, may have influenced the impact of the drugs on disease progression, considering the established efficacy of SGLT2is in these conditions [2,59]. However, in the RECOVERY trial, a post hoc subgroup analysis was performed between patients with and without diabetes, yielding insignificant results. Despite the given evidence, trials are unlikely to derive a definitive conclusion. Our analysis of hospital discharge within one month yielded a pooled risk ratio of 1.02 (CI: 1.00; 1.05), with the lower limit of the confidence interval intersecting 1, underscoring the necessity for further trials to establish a conclusive inference.

We performed a meta-analysis of eligible observational studies to further understand the effects of SGLT-2 inhibitors in COVID-19 patients with a history of SGLT-inhibitor use. Among the 17 observational studies analysed, 12 focused on mortality outcomes. These 12 studies collectively included 20,418 COVID-19 patients with prior SGLT2i use and 104,254 COVID-19 patients without prior SGLT2i exposure. The analysis revealed a significant 33% reduction in mortality odds among individuals with prior SGLT2i use compared to non-SGLT2i users. Except for the study by Kahkoska et al., all studies in the mortality analysis specifically included populations with pre-existing diabetes, potentially introducing bias with regard to our research question [33,34,35,36,38,39,41,42,43,44,47,49]. This suggests that the effects of medication could be attributed to its impact on diabetes rather than solely on COVID-19. However, drawing a definitive conclusion regarding this association remains challenging because of the inherent limitations of the available data.

Similarly, studies by Khunti et al., Shestakova et al., and Solerte et al. reported favourable outcomes in COVID-19 patients with T2DM taking SGLT2is [39,45,60]. Khunti et al.’s retrospective analysis involving nearly 3 million individuals showed a reduced COVID-19-related mortality risk (adjusted hazard ratio: 0.82) for SGLT2i users compared to those not on glucose-lowering medications [39,45,60]. The study of Shestakova et al. highlighted a significantly lower fatality rate among T2DM patients on SGLT2is before infection [45]. Solerte et al.’s recent research indicated lower death rates in COVID-19 patients with diabetes receiving SGLT2i therapy upon hospital admission [60].

A comparison of these medications provides insights into their relative efficacy. Shestakova et al.’s study showed varied mortality outcomes between different glucose-lowering therapies among patients with T2DM. Specifically, glucagon-like peptide-1 receptor agonists exhibited a lower case fatality rate (CFR) than SGLT2is [45]. In contrast, Israelsen et al. reported comparable 30-day mortality rates between users of GLP-1 receptor agonists (GLP-1 RAs) and SGLT-2 inhibitors (3.3% vs. 3.7%), and both were lower than those for DPP-4i users (8.6%) [34]. Khunti et al., however, concluded that there was little overall variation in outcomes for all glucose-lowering treatments, including SGLT2is, and that these differences were probably confounded by indication [39]. Our subgroup analysis confirmed that SGLT2is had a better outcome than GLP-1 RA or DPP-4i, which aligns with all observational studies analysed.

The assessment of hospitalisation risk in COVID-19 patients across observational studies revealed a 16% reduction in odds. When SGLT2is were directly compared with GLP-1 RA/DDP-i, there was a 17% decrease in odds. Israelsen et al., however, reported that hospital admission risks were generally comparable for GLP-1 RA and DPP-4i users and SGLT2i users [34]. However, their study had a smaller sample size than those analysed for this outcome. The meta-analysis of ICU admission and mechanical ventilation across studies yielded statistically insignificant results, indicating no significant differences in odds ratios between the various treatments. These findings suggest that, while SGLT2is offer a potential advantage in reducing the risk of hospitalisation, they do not seem to confer additional benefits concerning ICU admission. It is plausible that patients requiring ICU care might already be in a more critical health state, where improvement with these oral hypoglycaemic agents becomes less discernible or less effective. This implies that medication might be more influential in mitigating less severe COVID-19 cases.

Understanding the potential efficacy of SGLT2is in addressing COVID-19 infection and its complications is crucial. Despite their primary use in diabetes management, drugs such as dapagliflozin show additional beneficial properties beyond glucose control. These medications have pleiotropic effects, including anti-inflammatory actions, enhanced myocardial function, and improved oxygen delivery, as documented in previous studies [61,62]. SGLT2is’ role in COVID-19 can be understood through multiple mechanisms (Figure 8a). One of them that emerges prominently is their influence on cellular pH levels. The binding of SARS-CoV-2 to the angiotensin-converting enzyme 2 (ACE2) is facilitated in acidic environments [50]. SGLT2is indirectly regulate this pH balance, potentially reducing the viral load. Additionally, SGLT2is upregulate ACE2 receptors, increasing Ang 1–7 levels, which are protective against acute respiratory distress syndrome (ARDS) caused by COVID-19 [63,64,65].

SGLT2is also significantly reduce inflammation and oxidative stress, two key factors in COVID-19’s pathogenesis [66]. By lowering proinflammatory cytokines such as IL-6 and TNF-a, SGLT2is could mitigate the cytokine storm associated with severe COVID-19 cases [67]. Furthermore, these inhibitors impact myocardial and endothelial function, reducing the likelihood of complications such as myocarditis and heart failure [68,69]. SGLT2is also induce metabolic alterations that influence the disease. For instance, they decrease lactate production and modulate the activity of the lactate/H+-symporter and Na+/H+ exchanger (NHE), reducing cellular swelling and death [70,71]. This mechanism is important, because SARS-CoV-2 infection often leads to tissue hypoxia and heightened lactate production [72]. Of note, dapagliflozin has been shown to lower the incidence of new-onset diabetes in COVID-19 patients [73]. This finding holds significance, as new-onset diabetes is a recognised complication of COVID-19, possibly attributed to viral-induced damage to pancreatic cells [74,75]. Moreover, SGLT2is might have a protective role against COVID-19-related renal complications. Acute kidney injury (AKI) is a common complication in COVID-19 patients [76,77], and SGLT2is’ nephroprotective properties could prove beneficial in these cases [77,78,79,80]. Figure 8b shows an illustrative diagram depicting the potential mechanisms through which SGLT2is can benefit COVID-19 patients.

SGLT2is have been shown to have significant beneficial effects on circulating insulinemia and may also influence the course of COVID-19 through their impact on insulin levels and related metabolic processes. SGLT2is lower blood glucose levels by promoting renal glucose excretion rather than directly stimulating insulin secretion. This mechanism leads to decreased insulin levels in the circulation, as the body requires less insulin to effectively manage its blood glucose levels [81]. They have also been associated with weight loss and reductions in body fat, which contribute to an improved insulin sensitivity [82]. Furthermore, SGLT2is can enhance glucagon levels, which may help to balance insulin secretion [81]. Reductions in visceral fat are also closely linked to insulin resistance [83]. Elevated insulin levels can exacerbate inflammation, contributing to the cytokine storm observed in severe COVID-19 cases. By lowering insulin levels, SGLT2is may help to reduce inflammation and improve outcomes for patients suffering from COVID-19 [84]. Moreover, they may reduce adipose tissue inflammation, which is a known contributor to severe COVID-19 complications [84]. Improved glycaemic control can also enhance immune function and reduce the risk of severe disease progression, particularly in patients with pre-existing conditions like diabetes [81]. SGLT2is have been shown to provide cardiovascular protection and improve lung function by reducing interstitial lung oedema and enhancing oxygen utilisation, which can be particularly beneficial for COVID-19 patients who experience respiratory distress or hypoxia [83,84].

However, it is crucial to consider the balance between these advantages and the potential risks associated with SGLT2is. SGLT2is can increase the likelihood of diabetic ketoacidosis (DKA), especially regarding COVID-19 infection, which can trigger DKA [85]. Additionally, they may exacerbate low blood pressure in severe COVID-19 cases and increase the risk of urogenital fungal infections, particularly when combined with treatments such as dexamethasone and antibiotics commonly used in COVID-19 [86,87,88,89,90]. However, our analysis of diabetic ketoacidosis did not show a statistically significant association, suggesting that the observed risk of DKA with SGLT2is in the context of COVID-19 requires further investigation.

The umbrella review part of our study comprised the following ten reviews: one systematic review of case reports [51], six systematic reviews and meta-analyses [7,8,53,54,57,58], one meta-analysis [56], and two network meta-analyses [4,65]. The review population consisted of patients with T2DM who contracted COVID-19 and were on SGLT-2 inhibitors either before or after their hospitalisation or the onset of infection. The reviews primarily assessed outcomes such as the severity of COVID-19 infection [7,53,57], the need for Intensive Care Unit (ICU) admission, invasive and non-invasive mechanical ventilation [4,7,8,53,57], Disseminated Intravascular Coagulation (DIC) [57], Euglycemic Diabetic Ketoacidosis (Eu-DKA) [51,54,58], and in-hospital mortality [4,7,8,53,54,56,57,58,65].

The reviews suggested that the prior use of SGLT2is was associated with reduced adverse outcomes related to COVID-19 and notably lowered mortality rates among individuals with diabetes [7,8,54,56,57,65]. It was also noted that there was a decreased risk of hospitalisation [50] and a reduced risk of requiring non-invasive or invasive mechanical ventilation [69]. While a review of case reports suggested a possible increase in the risk of euglycemic diabetic ketoacidosis [51] with SGLT2is, the level of evidence was deemed non-significant. A systematic review and meta-analysis supported the notion that SGLT-2 inhibitors reduce mortality in COVID-19 without increasing the risk of DKA [54]. Conversely, another review found that prior SGLT-2 inhibitor treatment, when associated with euglycemic diabetic ketoacidosis, might protect renal function in COVID-19 patients with pre-existing ketotic states [58]. The reviews mostly fell into the Class II level of evidence, three were non-significant, and two were classified as having a Class IV level of evidence (Table 2). Our analysis of both observational studies and trials showed no significant increase in the odds of developing DKA. Based on this collective assessment, it appears that SGLT2is may not inherently lead to DKA.

Although SGLT2is such as dapagliflozin offer promising therapeutic avenues to manage COVID-19 and its complications, carefully considering their risks and benefits is essential. Their role in modulating pH levels, reducing inflammation and oxidative stress, and protecting against metabolic, cardiac, and renal complications positions them as valuable pharmacological agents for COVID-19 management. However, their utilisation requires careful consideration, particularly in severely ill COVID-19 patients.

## 5. Conclusions

The prophylactic use of SGLT2is reduces mortality rates and hospitalisation in COVID-19 patients, particularly in patients with diabetes. However, their utility as interventions after hospitalisation remains uncertain, highlighting the need for larger randomised controlled trials. The medication does not seem to have any significantly associated adverse events. The efficacy of SGLT2is appears to be limited when administered to critically ill patients. The presence of comorbidities and varying trial inclusion criteria underscores the need for more focused investigations. Despite their potential, individualised considerations are essential before integrating SGLT2is into COVID-19 management.

## Figures and Tables

**Figure 1 diseases-13-00067-f001:**
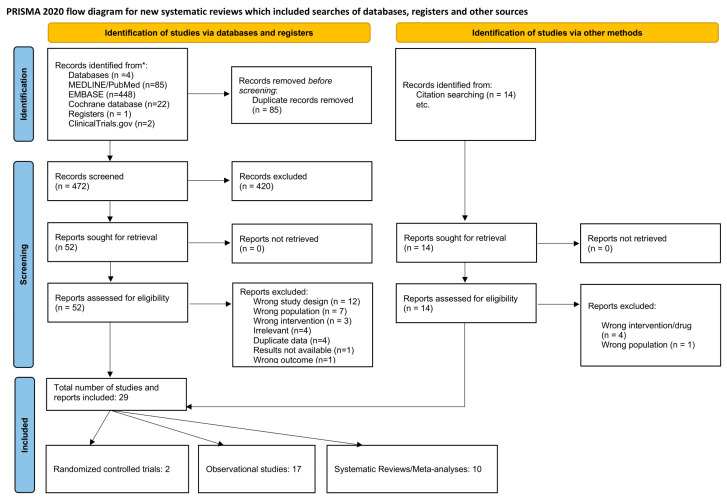
PRISMA flowchart outlining the screening process.

**Figure 2 diseases-13-00067-f002:**
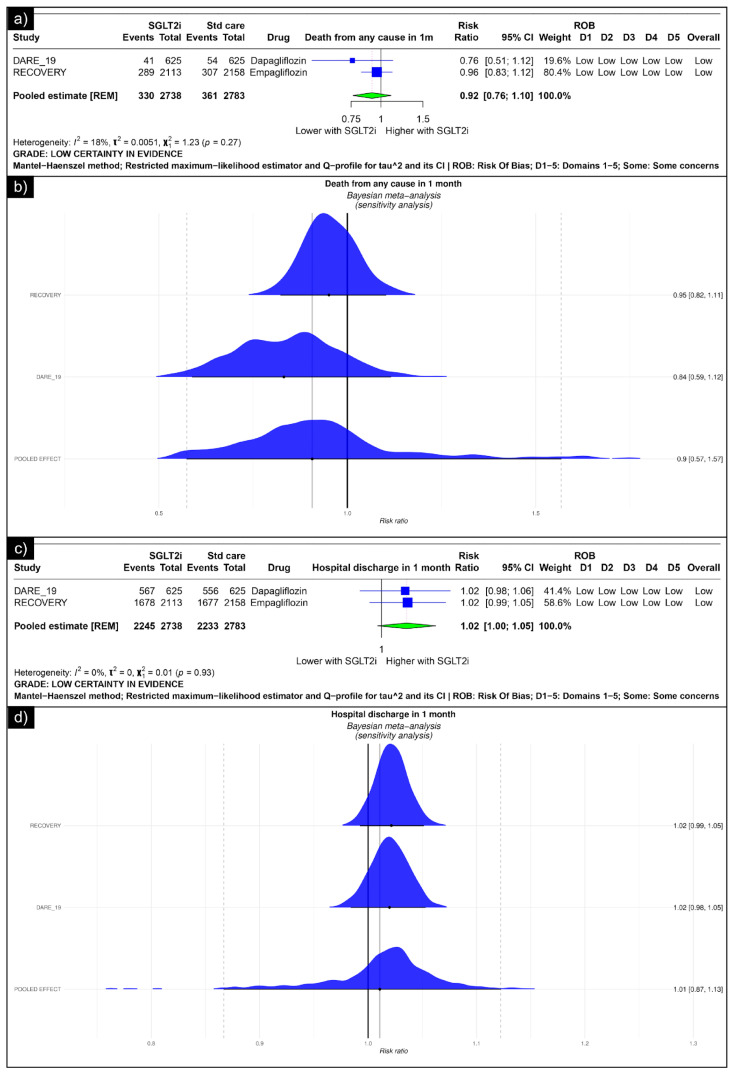
Forest plot of primary outcomes reported in trials: (**a**) Forest plot of “mortality” outcome reported in trials. (**b**) Bayesian Meta-Analysis (Sensitivity Analysis) of “mortality” outcomes reported in trials. (**c**) Forest plot of “hospitalisation” outcomes reported in the trials. (**d**) Bayesian Meta-Analysis (Sensitivity Analysis) of “hospitalisation” outcomes reported in the trials.

**Figure 3 diseases-13-00067-f003:**
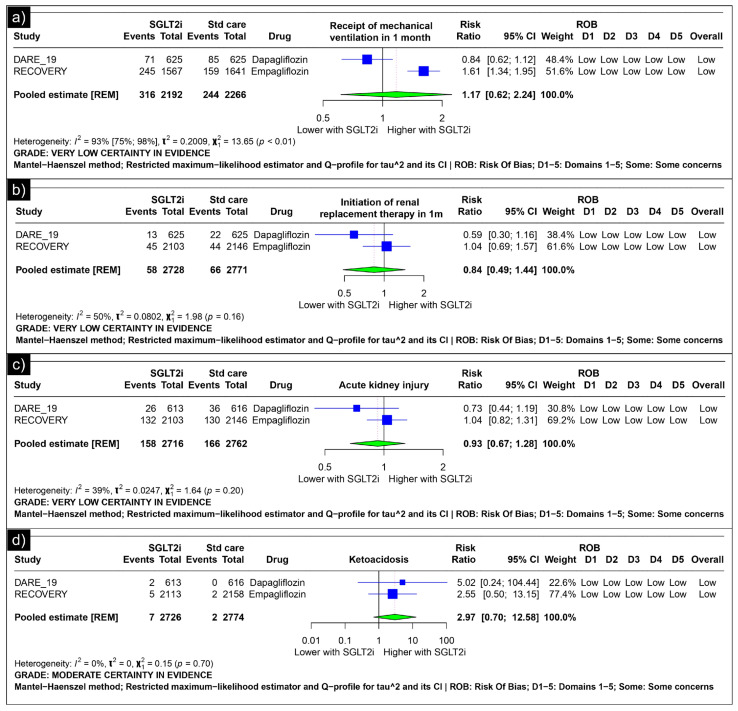
Forest plot of secondary outcomes reported in the trials. (**a**) Forest plot of “mechanical ventilation” outcomes reported in trials. (**b**) Forest plot of “renal replacement therapy” outcomes reported in trials. (**c**) Forest plot of “acute kidney injury” outcomes reported in trials. (**d**) Forest plot of “ketoacidosis” outcomes reported in trials.

**Figure 4 diseases-13-00067-f004:**
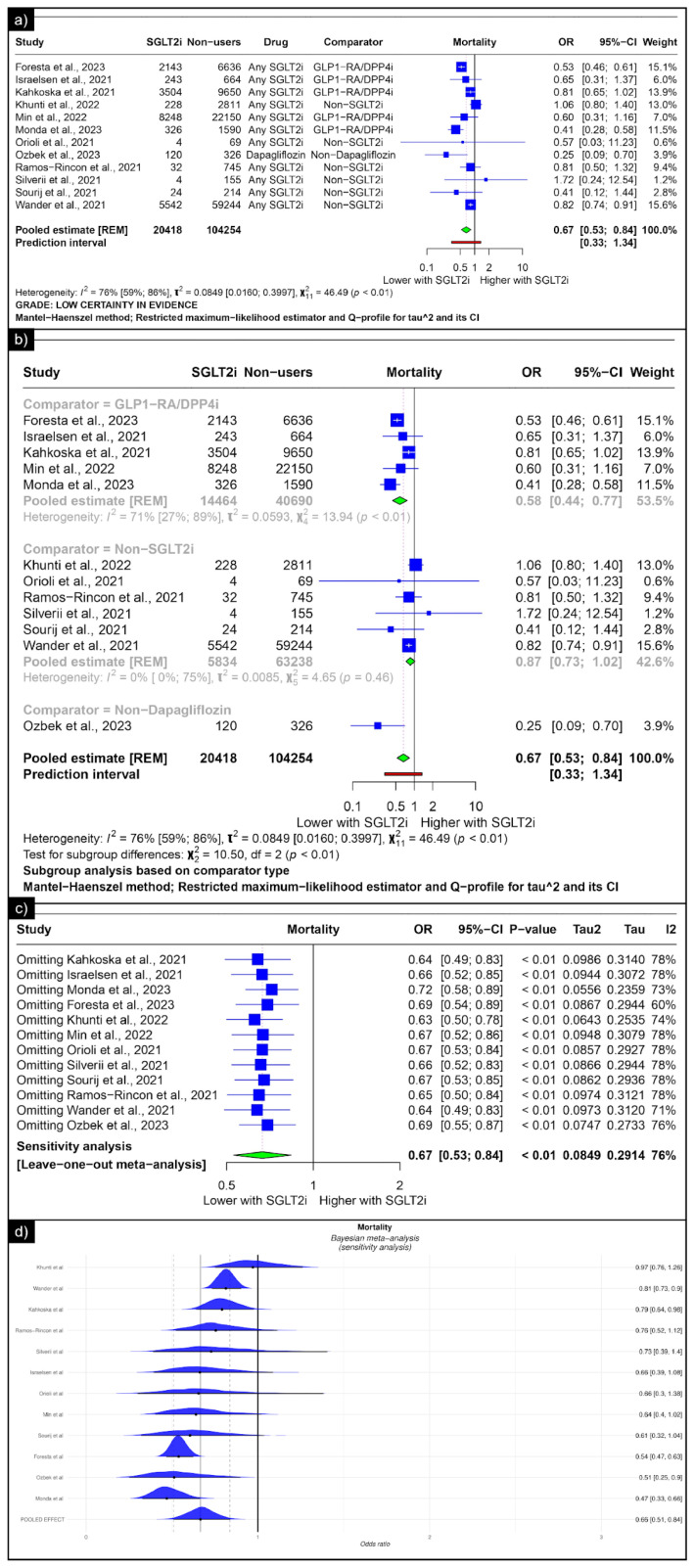
Forest plot of primary outcomes reported in observational studies: (**a**) Forest plot of “mortality” outcome reported in observational studies. (**b**) Sub-group analysis of “mortality” outcomes with SGLT2i use compared with GLP1-RA/DPP4i, non-SGLT2i users, and non-dapagliflozin users reported in observational studies. (**c**) Sensitivity Analysis (leave one out meta-analysis) of “mortality” outcome reported in observational studies. (**d**) Bayesian Meta-Analysis (Sensitivity Analysis) of “mortality” outcomes reported in observational studies [33,34,35,36,38,39,41,42,43,44,47,49].

**Figure 5 diseases-13-00067-f005:**
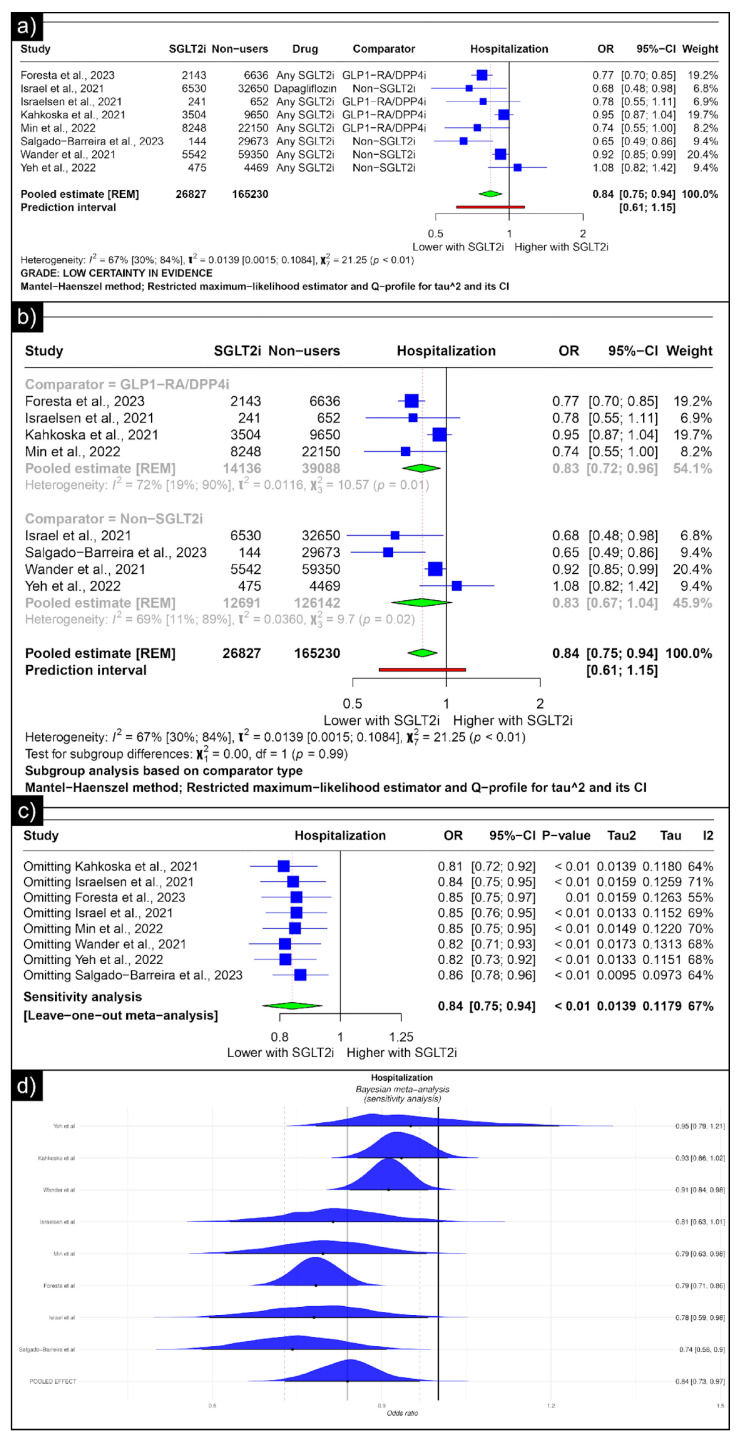
Forest plot of primary outcomes reported in observational studies: (**a**) Forest plot of “hospitalisation” outcome reported in observational studies. (**b**) Sub-group analysis of “hospitalisation” outcomes with SGLT2i use compared with GLP1-RA/DPP4i and non-SGLT2i use reported in observational studies. (**c**) Sensitivity Analysis (leave-one-out meta-analysis) of “hospitalisation” outcomes reported in observational studies. (**d**) Bayesian Meta-Analysis (Sensitivity Analysis) of “hospitalisation” outcomes reported in observational studies [33,34,37,38,40,41,44,48].

**Figure 6 diseases-13-00067-f006:**
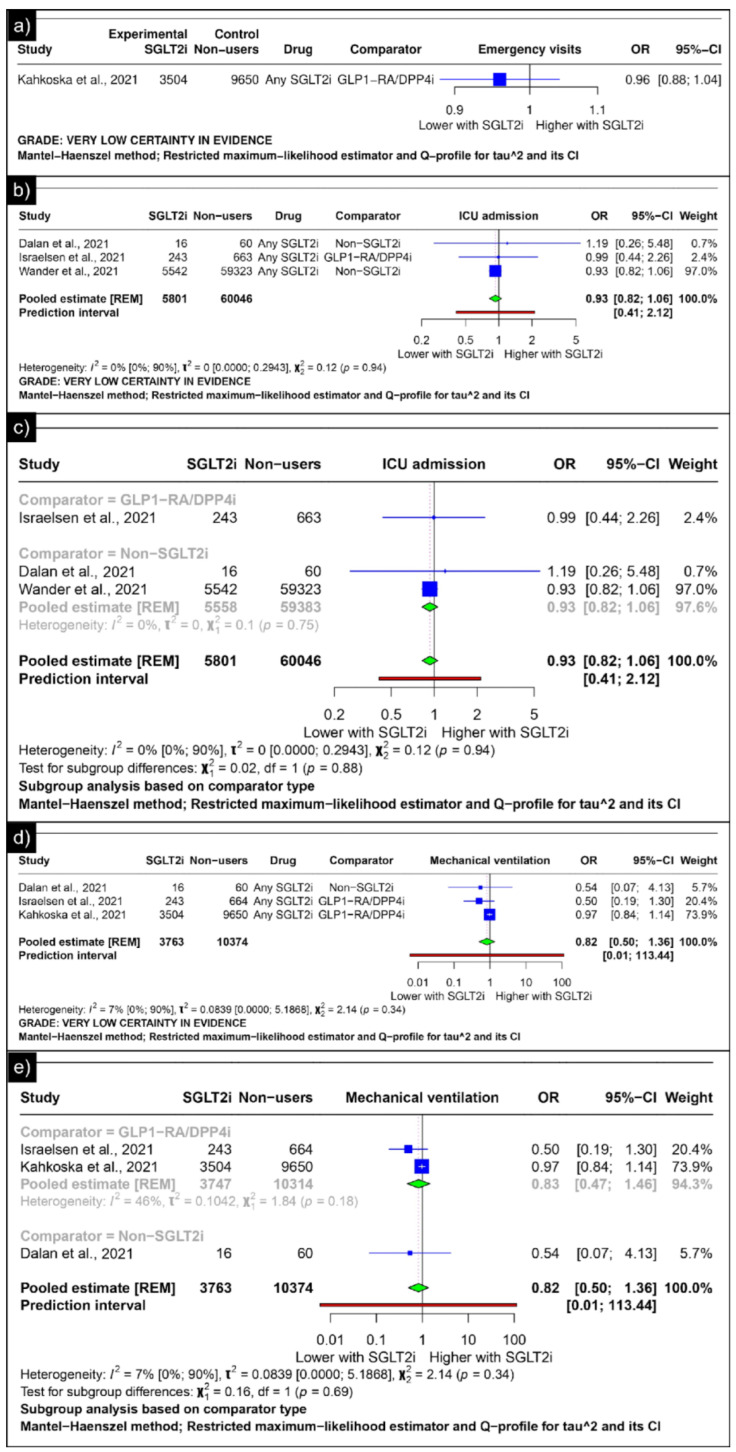
Forest plot of secondary outcomes reported in observational studies: (**a**) Forest plot of “emergency visits” outcome reported in observational studies. (**b**) Forest plot of “ICU admission” outcome reported in observational studies. (**c**) Subgroup analysis of “ICU admission” outcomes with SGLT2i use compared with GLP1-RA/DPP4i use and non-SGLT2i users reported in observational studies. (**d**) Forest plot of “mechanical ventilation” outcome reported in observational studies. (**e**) A sub-group analysis of the “mechanical ventilation” outcome with SGLT2i use was compared with GLP1-RA/DPP4i use, and non-SGLT2i users were reported in observational studies [33,34,44,46].

**Figure 7 diseases-13-00067-f007:**
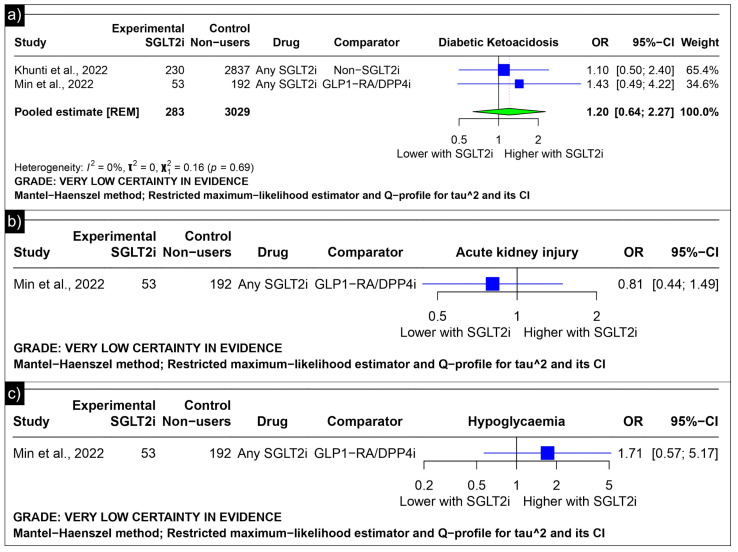
Forest plot of adverse outcomes reported in observational studies: (**a**) Forest plot of “Diabetic Ketoacidosis” outcome reported in observational studies. (**b**) Forest plot of “Acute Kidney Injury” outcome reported in observational studies. (**c**) Forest plot of “Hypoglycaemia” outcome reported in observational studies [39,41].

**Figure 8 diseases-13-00067-f008:**
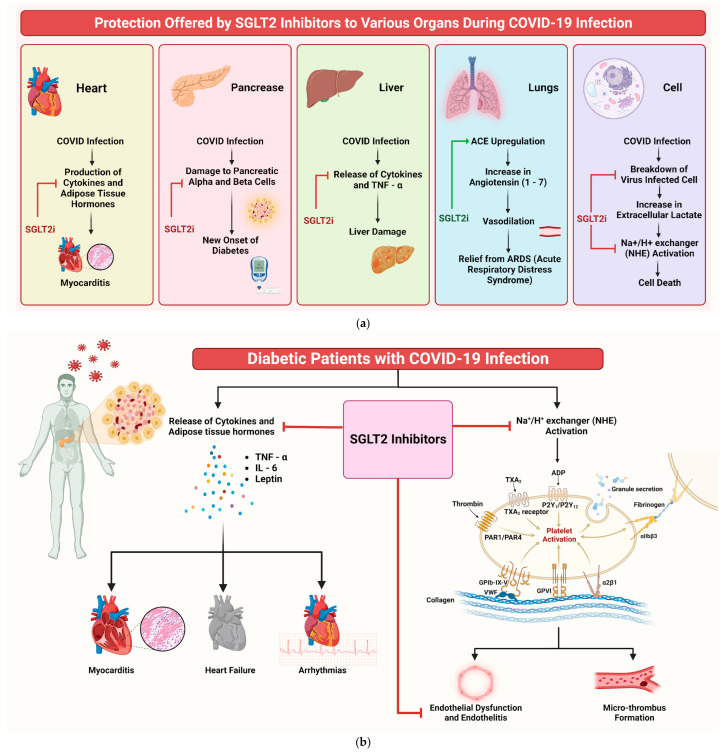
(**a**) Mechanism of protection offered by SGLT2is to various organs during COVID-19 infection. The red arrow indicates inhibition, while the green arrow indicates stimulation. (**b**) Infographic illustrating potential mechanisms of SGLT2is in benefiting COVID-19 patients.

**Table 1 diseases-13-00067-t001:** Population characteristics of the studies included in the review.

Title	Author and Year	Study Design	Population	Age	Sex	Intervention Arm	Comparator	Primary Outcomes	Conclusion	Quality Assessment
Empagliflozin in patients admitted to hospital with COVID-19 (RECOVERY): a randomised, controlled, open-label, platform trial.	RECOVERY Collaborative Group, 2023 [31]	Randomised Control Trial	Hospitalised patients with confirmed COVID-19 infection.	61.1 (mean age)	37% female, 63% male	Empaglifozin	Standard Care	All-cause mortality ^a^, discharge from hospital ^a^, invasive mechanical ventilation ^a^, and death ^a^.	Empagliflozin showed no impact on 28-day mortality, hospital stay, or the risk of progressing to invasive ventilation or death.	Low
Dapagliflozin in patients with cardiometabolic risk factors hospitalised with COVID-19 (DARE-19): a randomised, double-blind, placebo-controlled, phase 3 trial.	Kosiborod, 2021 [32]	Randomised Control Trial	Adults hospitalised with COVID-19 with at least one cardiometabolic risk factor.	61.4 (mean age)	42.6% female, 57.4% male	Dapagliflozin	Placebo	Time to new/worsened respiratory, cardiovascular, or kidney dysfunction during hospital stay, 30-day all-cause mortality ^a^, composite kidney outcome, days alive without mechanical ventilation, hospital discharge ^a^.	Dapagliflozin did not significantly reduce organ dysfunction or death but was well tolerated.	Low
Association Between Glucagon-Like Peptide 1 Receptor Agonist and Sodium–Glucose Cotransporter 2 Inhibitor Use and COVID-19 Outcomes.	Kahkoska, 2021 [33]	Retrospective Observational Study	COVID-19-positive adults with a GLP1-RA, SGLT2i, or DPP4i prescription within 24 months of a positive COVID-19 test.	58.6 ± 13.1 years	53.4% female, 46.6% male.	SGLT2i	GLP1-RA, DPP4i	60-day mortality, total mortality *^,b^, hospitalisation, and mechanical ventilation within 14 days.	Premorbid SGLT2i use was linked to lower mortality and adverse outcomes.	Serious
Comparable COVID-19 outcomes with current use of GLP-1 receptor agonists, DPP-4 inhibitors, or SGLT-2 inhibitors among patients with diabetes who tested positive for SARS-CoV-2.	Israelsen, 2021 [34]	Cohort Study	Adults infected with COVID-19 with a previous diagnosis of diabetes.	58–68 (Median age = 59)	38.2% female, 61.8% male.	SGLT2i	DPP4i, GLP1-RA	Hospital admission, ICU admission, mechanical ventilation, and 30-day mortality *^,b^.	GLP-1 RA and DPP-4i did not improve outcomes compared to SGLT2is in diabetes with SARS-CoV-2.	Serious
Protective Effects of Home T2DM Treatment with Glucagon-Like Peptide-1 Receptor Agonists and Sodium–Glucose Co-transporter-2 Inhibitors Against Intensive Care Unit Admission and Mortality in the Acute Phase of the COVID-19 Pandemic: A Retrospective Observational Study in Italy.	Monda, 2023 [35]	Retrospective Observational Study	Hospitalised COVID-19 patients with Type 2 DM with premorbid use of SGLT2is, Glp1-Ra, or Dpp-4i monotherapy.	61 ± 8 years	Male/female ratio = 0.9	SGLT2i	GLP1RA, DPP4i	Mortality rate *^,b^.	GLP-1ra and SGLT2i can be considered drugs of choice for patients with COVID-19 and diabetes.	Serious
The effect of dapagliflozin use on cardiovascular outcomes in type 2 diabetic patients hospitalised with COVID-19.	Ozbek, 2023 [36]	Case–Control Study	Patients with diabetes hospitalised with COVID-19.	64.32 ± 10.92 years	52.2% female, 47.8% male	Dapagliflozin	Other oral anti-diabetic medication (not specified).	Major adverse cardiovascular events, all-cause mortality *^,b^.	Dapagliflozin reduced mortality risk and ICU admission in patients with COVID-19 and diabetes.	Critical
Effect of dapagliflozin on COVID-19 infection and risk of hospitalisation.	Salgado-Barreira, 2023 [37]	Case–Control Study	Adults with PCR-positive COVID-19 infection.	59–84 (mean age 73)	Male/female ratio = 1	Dapagliflozin	NA	Hospitalisation ^b^, ICU admission, in-hospital death, and progression to severe COVID-19.	Dapagliflozin before SARS-CoV-2 infection did not increase risks.	Moderate
Dipeptidyl Peptidase-4 Inhibitors, Glucagon-like Peptide-1 Receptor Agonists, and Sodium–Glucose Cotransporter-2 Inhibitors and COVID-19 Outcomes.	Foresta, 2023 [38]	Retrospective Cohort Study	Adults > 40 years with at least two prescriptions of DPP-4i, GLP-1 RA, or SGLT2is or any other antihyperglycemic drug and a diagnosis of COVID-19.	Mean 71.86 years	40.3 female, 59.7 male	SGLT2i	DPP4i, GLP1-RA	Hospitalisation, in-hospital mortality, all-cause mortality *^,b^.	DPP-4i users had reduced COVID-19 mortality risk; GLP-1 RA and SGLT2is showed positive trends.	Serious
Association Between SGLT2 Inhibitor Treatment and Diabetic Ketoacidosis and Mortality in People With Type 2 Diabetes Admitted to Hospital With COVID-19.	Khunti, 2022 [39]	Retrospective Cohort Study	Adults with Type 2 DM who were admitted to the hospital with COVID-19 infection.	72 years (mean age)	37.7% female, 62.3% male	SGLT2i	NA	DKA development, death in different cohorts *^,b^.	SGLT2i prescription had low DKA risk and no increased in-hospital mortality in T2D with COVID-19.	Serious
Identification of drugs associated with reduced severity of COVID-19: a case–control study in a large population.	Israel, 2021 [40]	Case–Control Study	Adults hospitalised with COVID-19 infection.	64.6 (mean age)	Male/female ratio = 1	NA	NA	Hospital admission ^b^.	Antidiabetic drugs showed protective effects against COVID-19 severity.	Moderate
Association between antidiabetic drug use and the risk of COVID-19 hospitalisation in the INSIGHT Clinical Research Network in New York City.	Min, 2022 [41]	Retrospective Cohort Study	Adult patients in the INSIGHT CRN with evidence of T2D and at least one HbA1c and serum creatinine measurement in the year prior to the index date of 15 March 2020.	62 ± 13 years	Female = 51%, Male = 49%	SGLT2i	DPP4i, GLP1-RA, sulphonylureas	Risk of COVID-19 hospitalisation ^b^, AKI, DKA, hypoglycaemia, in-hospital death *^,b^.	Sulphonylureas, DPP-4 inhibitors, and GLP-1 agonists increased COVID-19 hospitalisation risk in T2D with metformin.	Moderate
Cardiometabolic therapy and mortality in very old patients with diabetes hospitalised due to COVID-19.	Ramos-Rincón, 2021 [42]	Observational Study	Patients ≥80 years with T2DM were hospitalised for COVID-19 between March 1 and May 29, 2020.	Mean age 86 (82.7–88.9)	Female = 47.1%, Male = 52.9%	DDP4i	ARB	In-hospital mortality *^,b^.	DPP-4 inhibitors and angiotensin receptor blockers were protective; acetylsalicylic acid increased in-hospital mortality.	Serious
Clinical characteristics and short-term prognosis of in-patients with diabetes and COVID-19: a retrospective study from an academic centre in Belgium.	Orioli, 2021 [43]	Retrospective Study	Patients with known or newly diagnosed diabetes and confirmed COVID-19.	The mean age was 69 (±14) years	Female = 52.0%, Male = 48.0%	NA	NA	Predictive factors of in-hospital death, ICU admissions, deaths *^,b^.	Diabetes comorbidities did not adversely affect mortality; confirmation is needed in a larger series.	Serious
Prior Glucose-Lowering Medication Use and 30-day Outcomes Among 64,892 Veterans With Diabetes and COVID-19.	Wander, 2021 [44]	Retrospective Study	Patients with diabetes and one or more positive nasal swab(s) for severe acute respiratory syndrome coronavirus 2 (1 March 2020–10 March 2021) (*n* = 64,892).	Mean age of 67.7	Female = 6.0%, Male = 94.0%	SGLT2i	GLP1-RA, ARB, Insulin	Associations of HbA1c and glucose-lowering medication use with hospitalisation, intensive care unit (ICU) admission, and mortality at 30 days *^,b^.	Higher HbA1c and insulin use linked to adverse outcomes; GLP1-RA, metformin, and SGLT2is were protective.	Serious
Risk factors for COVID-19 case fatality rate in people with type 1 and type 2 diabetes mellitus: a nationwide retrospective cohort study of 235,248 patients in the Russian Federation.	Shestakova, 2022 [45]	Retrospective Cohort study	Patients with DM of the National Diabetes Register (NDR) with a reported outcome of coronavirus infection.	NA	32% Males and 68% Females.	SGLT2i	Other Anti-Hypoglycaemic Medications (Metformin, DPP4i, GLP1-RA, etc.)	Association of demographic, clinical, and laboratory characteristics, pre-COVID-19 glucose-lowering therapy (in T2DM), and anti-COVID-19 vaccination status with the fatality cases due to COVID-19 and identify the risk factors for the death.	COVID-19 fatality risk increased with male gender, older age, longer DM duration; positive effects with certain glucose-lowering therapies.	Serious
The association of hypertension and diabetes pharmacotherapy with COVID-19 severity and immune signatures: an observational study	Dalan, 2021 [46]	Observational study	Patients with PCR confirmed COVID-19 who were hospitalised at the National Centre of Infectious Diseases (NCID), Singapore, up to 15 April 2020.	NA	NA	SGLT2i	DPP4i	Hypoxia (requirement for supplemental oxygen to maintain blood oxygen saturations >93%), intensive care unit (ICU) admission ^b^, mechanical ventilation, and death.	Health complications in hypertension and T2DM; diabetes and hypertension linked to hypoxia and ICU admission. ACE-I in hypertensive patients lowered ICU admission risk; ARBs increased it. DPP4i in patients with diabetes increased ICU risk, while SGLT2id reduced mechanical ventilation risk. ARB use is linked to higher inflammation; DPP4i is linked to lower neurotrophic factor levels.	Serious
Are diabetes and its medications risk factors for the development of COVID-19? Data from a population-based study in Sicily.	Silverii, 2020 [47]	Retrospective Observational Study	Patients with SARS-CoV-2 positive cases and deaths in Sicily region, up to 2020, May 14th.	The mean age was 73.31 (±12.66) years	Female = 45.9%, Male = 54.1%	NA	NA	Association of diabetes with COVID-19 prevalence, case fatality *^,b^, and the association between different diabetes medications and risk for COVID-19 infection and death.	DM was not a risk factor for COVID-19 infection; instead, it was associated with a higher case fatality.	Serious
Hospitalisation and mortality in patients with COVID-19 with or at risk of type 2 diabetes: data from five health systems in Pennsylvania and Maryland.	Yeh, 2022 [48]	Retrospective Cohort Study	Adult patients with T2DM or at risk of T2DM using the PaTH Toward a Learning Health System (PaTH) clinical data research network.	Mean age was 62.3 (±14.0) years	Female = 53.8%, Male = 46.2%	NA	NA	COVID-19 severity, phospitalisationb;, admission to the ICU, intubation, or death.	T2DM and insulin use increased ICU/intubation/death odds; metformin lowered odds. GLP-1 RA, DPP-4i, and metformin users had lower hospitalisation odds. Non-Hispanic Black and Hispanic diabetes-risk patients had higher hospitalisation odds. Later COVID-19 diagnoses were associated with lower odds of severe outcomes.	Serious
COVID-19 fatality prediction in people with diabetes and prediabetes using a simple score upon hospital admission	Sourij, 2021 [49]	Retrospective Cohort Study	People aged 18 years or older with a confirmed positive throat swab for SARS-CoV-2 and a confirmed diagnosis of type 1 diabetes, type 2 diabetes, or prediabetes were included in the registry (either known or newly diagnosed).	Mean age of people hospitalised (*n* = 238) for COVID-19 was 71.1 ± 12.9 years	Females = 36.4%, Males = 63.6%	NA	NA	In-hospital mortality in patients with diabetes and confirmed diagnosis of COVID-19 *^,b^.	The in-hospital mortality for COVID-19 was high in people with diabetes but not significantly different from the risk in people with prediabetes.	Serious

* = Outcomes included for meta-analysis; ^a^ = Outcome measure was pooled as risk ratio; ^b^ = Outcome measure was pooled as odds ratio.

**Table 2 diseases-13-00067-t002:** Characteristics of the included systematic reviews and meta-analyses.

Title	Author and Year	Type of Review	Population	Outcome	Conclusion	Class of Evidence	Reason for Classification
Association of Glucose-Lowering Drugs With Outcomes in Patients With Diabetes Before Hospitalisation for COVID-19: A Systematic Review and Network Meta-analysis.	Zhu Z, 2022 [8]	Systematic review and network meta-analysis	Patients with diabetes while receiving glucose-lowering therapies for at least 14 days before the confirmation of COVID-19 infection.	Need for intensive care unit admission, invasive and noninvasive mechanical ventilation, and in-hospital death.	Use of an SGLT2i before COVID-19 infection is associated with lower COVID-19-related adverse outcomes	Class II	Pooled Sample size: >1000) *p*-value: Significant differences reported as credible intervals. *I*^2^ (Heterogeneity): <50% Prediction Interval: Not explicitly reported Small-study effects: Funnel plots appear symmetric, indicating no small-study effects. Excess significance bias: Not observed in sensitivity analysis or publication bias assessments. Largest study: (Khunti, K. 2022) has a statistically significant effect size [39]
The Association Between Antidiabetic Agents and Clinical Outcomes of COVID-19 Patients With Diabetes: A Bayesian Network Meta-Analysis.	Chen Y, 2022 [50]	Bayesian Network Meta-analysis	Patients with COVID-19 and diabetes receiving specific antidiabetic agents.	Metformin, DPP4i, SGLT2i (OR, 0.82; *p* = 0.001), and glucagon-like peptide-1 receptor agonist (GLP1RA) treatments were associated with lower COVID-19 mortality in individuals with diabetes compared to respective nonusers.	Metformin, DPP4i, SGLT2i, and GLP1RA treatments were highly possible to reduced COVID-19 mortality risk in individuals with diabetes	Class II	Pooled sample size: >1000 *p*-value: 0.00 indicated. *I*^2^ (Heterogeneity): <50% Prediction Interval: Not explicitly reported Small-study effects: The trim-and-fill analysis suggested no difference Excess significance bias: Sensitivity analysis not conducted Largest study: Wander PL 2021 has a statistically significant effect size [44]
Sodium–glucose cotransporter-2 inhibitor-associated euglycemic diabetic ketoacidosis in COVID-19-infected patients: A systematic review of case reports	Khedr, A., 2023 [51]	Systematic review of case reports	SGLT2is in coronavirus disease 2019 (COVID-19) in patients with diabetes developing DKA.	Clinical manifestations, treatment approaches, and outcomes of eu-DKA in patients with concurrent COVID-19 infection.	SGLT2is may increase the risk of euglycemic diabetic ketoacidosis in COVID-19-infected patients with diabetes.	Non-significant	Not applicable. No quantitative analysis was performed.
Effect of Antidiabetic Therapy on Clinical Outcomes of COVID-19 Patients With Type 2 Diabetes: A Systematic Review and Meta-Analysis	Zhan, K., 2023 [52]	Systematic Review and Meta-analysis	COVID-19 patients with T2D taking one of the anti-diabetes drugs.	COVID-19-related deaths and other poor clinical outcomes, including ICU admission and invasive mechanical ventilation.	Usage of metformin, SGLT2is, and GLP-1ra could significantly decrease mortality in COVID-19 patients with T2D.	Class II	Pooled sample size: >1000 *p*-value: Statistically significant as inferred by 95% confidence intervals *I*^2^ (Heterogeneity): <50% Prediction Interval: Not explicitly reported Small-study effects: The trim-and-fill analysis suggested no difference (Egger’s *p*-value: 0.81) Excess significance bias: Sensitivity analysis not conducted Largest study: (Khunti, K. 2022) has a statistically significant effect size [39]
Noninsulin-based antihyperglycemic medications in patients with diabetes and COVID-19: A systematic review and meta-analysis	Nassar, M., 2023 [53]	Systematic Review and Meta-analysis	Patients with T2DM who were infected with COVID-19.	Effect of different AGMs on mortality, hospitalisation, and admission to the ICU and/or mechanical ventilation.	There was a decrease in the risk of hospitalisation with SGLT2is.	Non-significant	Pooled sample size: >1000 *p*-value: Non-significant
Pre-admission use of sodium–glucose transporter-2 inhibitors (SGLT-2is) may significantly improve COVID-19 outcomes in patients with diabetes: A systematic review, meta-analysis, and meta-regression	Permana, H., 2023 [54]	Systematic review, meta-analysis, and meta-regression	SGLT2i in patients with diabetes and COVID-19.	Mortality from COVID-19, severe COVID-19, and diabetic ketoacidosis (DKA).	The use of SGLT2is as glucose-lowering treatment in patients with diabetes may reduce mortality and severity of COVID-19 but without increased risk of developing diabetic ketoacidosis (DKA).	Class II	Pooled sample size: >1000 *p*-value: 0.00001 (mortality) *I*^2^ (Heterogeneity): >50% Prediction Interval: Not explicitly reported Small-study effects: The trim-and-fill analysis suggested no difference (symmetrical). Egger’s *p*-value: 0.13 Excess significance bias: Sensitivity analysis not performed Largest study: (Khunti, K. 2022) has a statistically significant effect size [39]
Effects of Novel glucose-lowering Drugs on COVID-19 Patients with Diabetes: A Network Meta-analysis of Clinical Outcomes	Yang, Y., 2024 [55]	Network meta-analysis	Adult (aged ≥ 18 years) patients with COVID-19 and diabetes. Intervention measures are novel glucose-lowering drugs (e.g., SGLT2is, GLP-1RA, and DPP4i).	The primary outcome was the mortality of people with diabetes and COVID-19, and the secondary outcomes were required intensive care unit (ICU) admission and mechanical ventilation.	SGLT2is are associated with a lower mortality rate in people with diabetes and COVID-19. SGLT2is are linked to lower mechanical ventilation requirements.	Class II	Pooled sample size: >1000 *p*-value: The exact value cannot be ascertained. Study claims statistical significance Loop specific heterogeneity τ^2^ = 0.127. Prediction Interval: Not explicitly reported Small-study effects: Presence of publication bias. Excess significance bias: Sensitivity analysis not conducted Largest study: Kahkoska 2021 had a statistically significant effect size [33]
Preadmission use of antidiabetic medications and mortality among patients with COVID-19 having type 2 diabetes: A meta-analysis	Nguyen, N.N., 2022 [56]	Meta-analysis	Patients with confirmed COVID-19 who had diabetes and were on prehospital medications extending to the pandemic.	The primary outcome was in-hospital mortality or mortality within 90 days.	Metformin, GLP-1RA, and SGLT2is were associated with lower mortality rates in patients with COVID-19 having T2DM.	Class IV	Pooled sample size: >1000 *p*-value: <0.01 *I*^2^ (Heterogeneity): >50% Prediction Interval: Not explicitly reported Small-study effects: No publication bias was found using Egger’s test Excess significance bias: Sensitivity analysis was insignificant Largest study: (Khunti, K. 2022) has a statistically significant effect size [39]
Association Between Anti-diabetic Agents and Clinical Outcomes of COVID-19 in Patients with Diabetes: A Systematic Review and Meta-Analysis	Han, T., 2022 [57]	Systematic Review and Meta-Analysis	Patients were older than 18 years of age and diagnosed with both diabetes and COVID-19; Home use or in-hospital use of specific anti-diabetic agents.	Clinically validated definition of death, poor composite outcomes comprising intubation ventilation, Acute Respiratory Distress Syndrome (ARDS), disseminated intravascular coagulation (DIC), intensive care unit (ICU) admission, disease progression, or other adverse outcomes.	Metformin might be beneficial in decreasing mortality and poor composite outcomes in patients with diabetes and infected with SARS-CoV-2. DPP-4 inhibitors, sulfonylurea/glinides, SGLT-2 inhibitors, and GLP-1RA would not be adverse.	Non-significant	Pooled sample size: <1000 *p*-value: 0.904
Diabetes-related acute metabolic emergencies in COVID-19 patients: a systematic review and meta-analysis	Papadopoulos, V.P., 2021 [58]	Systematic review and meta-analysis	COVID-19 patients who had developed either DKA, HHS, or combined DKA/HHS, or EDKA before administration of antidiabetic treatment, including insulin, metformin, sulfonylureas, DPP-4, glucagon-like peptide-1 receptor agonists (GLP-1 RAs), SGLT2i, and pioglitazone.	Primary (survival/discharge vs. death) and secondary (type of metabolic emergency) outcomes concerning origin, the coexistence of ketotic, and hyperosmotic state.	Previous SGLT2i treatment, though associated with EDKA, might preserve renal function in COVID-19 patients.	Class IV	Pooled sample size: <1000 *p*-value: 0.004 *I*^2^ (Heterogeneity): Not specified for SGLTi outcome Prediction Interval: Not explicitly reported

## Data Availability

The data supporting the findings of this article are available from the corresponding author upon reasonable request.

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
