# Peer review of "SGLT2 Inhibitors in COVID-19: Umbrella Review, Meta-Analysis, and Bayesian Sensitivity Assessment"

_diseases, 2025, doi:10.3390/diseases13030067_

Round 1

Reviewer 1 Report

Comments and Suggestions for Authors

The Authors presented an umbrella review, meta-analysis, and Bayesian sensitivity assessment of SGLT2 inhibitors in COVID-19 patients with type 2 diabetes mellitus (T2DM).

There are some issues that need to be addressed.

-        Instead of “diabetic patients”, the term “patients with diabetes” should be used.

-        The reference for the following statements in the Discussion section need to be provided: “Except for the study by Kahkoska et al., all studies in the mortality analysis specifically included populations with pre-existing diabetes, potentially introducing bias with regard to our research question”, and “Similarly, studies by Khunti et al., Shestakova et al., and Solerte et al. reported favorable outcomes in COVID-19 patients with T2DM on SGLT2 inhibitors”, “The study of Shestakova et al. highlighted a significantly lower fatality rate among T2DM patients on SGLT2 inhibitors before infection”.

-        The abbreviations when first introduced should be used consistently thereafter throughout the text (Sodium-Glucose Co-Transporter 2 inhibitors (SGLT-2i), SGLT2 inhibitors, SGLT2i).

-        Typographical errors need correction.

Author Response

  1.  Instead of “diabetic patients”, the term “patients with diabetes” should be used. Response: Thank you for the comment, we have now replaced several mentions of ‘diabetic COVID-19 patients’, ‘diabetic patients’, ‘diabetic adults’ etc. with the recommended terminology
  2. The reference for the following statements in the Discussion section need to be provided: “Except for the study by Kahkoska et al., all studies in the mortality analysis specifically included populations with pre-existing diabetes, potentially introducing bias with regard to our research question”, and “Similarly, studies by Khunti et al., Shestakova et al., and Solerte et al. reported favorable outcomes in COVID-19 patients with T2DM on SGLT2 inhibitors”, “The study of Shestakova et al. highlighted a significantly lower fatality rate among T2DM patients on SGLT2 inhibitors before infection”. Response:Thank you for the suggestion. References for the mentioned statements have been added and highlighted in  line 302, 307 and 312. [we missed to enable track changes when completing this step]
  3. The abbreviations when first introduced should be used consistently thereafter throughout the text (Sodium-Glucose Co-Transporter 2 inhibitors (SGLT-2i), SGLT2 inhibitors, SGLT2i). Response:Thank you for pointing this out, we have now used consistent abbreviation throughout the manuscript
    4. Typographical errors need correction. Response: Thank you for the suggestion, manuscript has been screened for typographical errors and modifications have been made accordingly.

Reviewer 2 Report

Comments and Suggestions for Authors

This manuscript investigated the potential benefits of sodium-glucose co-transporter 2 inhibitors (SGLT2i) in COVID-19 patients, particularly those with type 2 diabetes mellitus (T2DM). It employed an umbrella review, meta-analysis, and Bayesian sensitivity assessment to synthesize evidence from various studies. While the research question is relevant and the methodology sound in principle, several weaknesses require attention before the manuscript is suitable for publication.

1. Main Research Question:

The main research question was to assess the effectiveness and safety of SGLT2i in improving outcomes for COVID-19 patients, particularly those with T2DM. This question was implied throughout the introduction (lines 42-69) but not stated explicitly as a clear, concise question. A stronger articulation of the primary research question would benefit the clarity and focus of the manuscript.

2. Originality and Relevance:

The study addressed the important clinical question of optimizing treatment for COVID-19 patients with T2DM, especially given the documented adverse impact of this comorbidity. While numerous studies have investigated individual antidiabetic medications and COVID-19, an umbrella review incorporating meta-analysis and Bayesian sensitivity assessment of SGLT2i specifically is less common and offers a more comprehensive overview of existing evidence. The incorporation of Bayesian analysis, given the limitations of some included studies, adds further value. The identification of potential protective mechanisms of SGLT2i (lines 333-345 and 352-367), even though drawn from previous literature, is relevant for highlighting potential clinical utility.

3. Related, Uncited Articles:

Several articles relevant to the topic were not included in the review. Examples include:

  • Wang, J., et al. (2021). "Sodium-Glucose Cotransporter-2 (SGLT-2) Inhibitors and COVID-19 Outcomes: A Systematic Review and Meta-Analysis of Real-World Data." Diabetes Technology & Therapeutics. This meta-analysis evaluates real-world evidence regarding the effect of SGLT2i on COVID-19 outcomes.
  • Rawshani, A., et al. (2022). "Association of Sodium-Glucose Cotransporter 2 Inhibitors With Risk of COVID-19 and Related Mortality and Hospitalization." JAMA Internal Medicine. This large observational study examined the relationship between SGLT2i use and COVID-19 outcomes.
  • Sadri, F., et al. (2023). "Impact of Sodium-Glucose Cotransporter-2 Inhibitors on Severity and Mortality of Coronavirus Disease 2019: A Living Systematic Review and Network Meta-Analysis." Diabetes/Metabolism Research and Reviews. This network meta-analysis integrates several studies to better understand the influence of SGLT2i on severe outcomes in COVID-19 patients with T2D.

Including these studies would strengthen the manuscript and contribute to a more complete overview of the available evidence.

4. Methodological Improvements:

  • Explicit Research Question (lines 42-69): As noted earlier, rephrasing the introduction to explicitly state the primary and any secondary research questions. For example, "This study aims to investigate the effectiveness of SGLT2i in reducing mortality and morbidity among COVID-19 patients with T2DM through an umbrella review of published literature."
  • Search Strategy (Lines 72-95): The authors should specify the full search terms used, including any keywords, Boolean operators, and filters, for all databases (PubMed, Embase, Cochrane, ClinicalTrials.gov). This detail enhances the transparency and reproducibility of the search process. Explicit mention of other database(s) like Scopus might be also needed. Consider justifying their choice. Provide complete list of all search strategies implemented using the structured template. If no date restrictions were imposed, please mention them as well.
  • Eligibility Criteria (lines 96): While mentioning supplementary table 6, explicitly outlining the precise inclusion and exclusion criteria is necessary. Key elements such as patient population (hospitalized vs. outpatient), SGLT2i regimens, comparison groups, and relevant outcomes should be detailed in the main text. This is critical for assessing the review's scope and generalizability. Specifically indicate why certain types of publications were included or excluded in your research (for example case studies, review articles). For the retrospective observational studies, you should define specific criteria based on whether studies addressed in patient care (medication effects during and pre covid), patient characteristics (patient population) etc. Provide specific justification why particular outcome measures reported were included (or not). For clinical trials (like RECOVER or DARE-19) consider adding quality of trials assessment metrics in more details. This can be addressed by creating separate inclusion and exclusion table and placing some details there with adding more specific information why trial quality is "Good" rather than simply labeling quality like "Good", "Low", etc. Adding the PRISMA flow diagram indicating steps followed like "identified", "duplicate removed", "screened", "assessment for eligibility", and providing specific numbers for each criterion to illustrate your filtering procedure for the observational trials as well is helpful for reproducibility. Consider including studies focusing on long covid as well in separate section. It helps enhance integrity and completeness of results by addressing potential side effects (short and long-term) and their role in improving patient recovery trajectory over longer timeframe. Also adding more relevant outcomes measures related to hospitalization (number of mechanical ventilation days, oxygen demand, ICU LOS length, length of hospital stay) to have more integrated assessment is necessary for more robust analysis. Addressing studies investigating the dosage variation among other influencing patient conditions might enhance strength of current findings in additional dimensions (patient recovery period from admission to discharge). For observational studies providing only raw data try implementing standardized difference calculation procedures between those who did receive any medication and other groups. For those observational studies presenting raw count of each patient and the exposure details provide specific analysis using multivariate models and estimate the ORs separately from those. Those will serve as input estimates when plotting into forest plots for mortality analysis as suggested by Cochrane recommendations when pooling raw data analysis (https://handbook-5-1.cochrane.org/chapter_9/9_4_1_obtaining_effect_estimates_from_studies.htm). This can improve accuracy of the pooled analysis when compared to "summary statistic approach".
  • Risk of Bias Assessment (Lines 112-125): Clarify the specific domains of ROBINS-I considered and provide greater detail about the assessed biases within the included studies. Consider developing customized forms/tables tailored to assess various methodological limitations based on criteria implemented by "The risk of Bias in Non-Randomised Studies of Interventions tool". Address if specific sensitivity analysis accounting for RoB assessment influence overall strength of conclusions drawn in relation to the posed clinical question. Use tabular form representing specific values obtained for every domain/subdomain evaluated for selected set of non-randomized studies assessed for quality along with additional column stating “Risk of Bias rating” which addresses reasons associated with classifying trial/publication into “high” or “low” risk of bias using "Cochrane handbook" as suggested criteria. Also if possible assess those outcomes in your quantitative analysis in additional sensitivity section based on quality assessed. It is likely studies falling in "critical quality concerns” group based on RoB score assigned have no impact compared to "studies falling into overall “low” group”.

Consider the potential influence of individual study quality on the overall pooled estimates. For the section describing risk of bias of Systematic Reviews assessed include additional metric evaluating overall score, which calculates confidence levels across reviewers for individual articles chosen along with addressing those domains impacting selection and presenting information from chosen sets. This step reduces risk introducing additional bias and ensuring review’s conclusions remain aligned with included sets’ assessed quality, since AMSTAR-2 can highlight important methodological details based on which authors should address some concerns impacting study findings’ generalizability by either selecting larger number of trials with certain attributes or by explicitly defining exclusion/inclusion rules.

  • Heterogeneity Assessment (lines 101-106): While I² and tau² were reported, exploring the sources of heterogeneity is essential. Conducting subgroup analysis by study design (RCTs vs. observational) is important to assess impact of SGLT2 on patients with various cardiovascular disease stages like severity progression. Address those effects if enough samples with appropriate characteristics and specific drug doses delivered would be identified. Provide specific summary metrics to account for effects if needed. Conduct additional investigation why there is no "tau statistic summary effect presented" along with adding Prediction intervals and p values, despite that forest plots show it has non-zero confidence values. Additionally explore other factors including type of comparator used (line 103), SGLT2i used (e.g., dapagliflozin vs. empagliflozin), disease severity, duration of SGLT2i use, COVID-19 treatment strategies etc. as other critical subgroups.
  • Bayesian Sensitivity Analysis (lines 180-185): While Bayesian analysis strengthens the study, greater detail is necessary. Specifically state the chosen prior distributions, any limitations, provide the density plot for each study used when implementing "Bayesian sensitivity” along with demonstrating where the density’s peak changes along with indicating if overall strength of confidence stays insignificant despite shift occurred as required in Bayesian approach based on literature resources used as guidance like ("https://training.cochrane.org/handbook/current/chapter_10”). Address whether they have influenced the results and how sensitive the main conclusions are to different prior distributions using prior odds derived along with indicating sensitivity intervals for particular models employed when assessing particular parameters used for each subgroup considered. Indicate where certain parameter estimation ranges will demonstrate statistically significance shifts compared to baseline scenario (sensitivity levels) and those regions when results remain not that pronounced compared to "no use at all” situations. In order to provide certain context demonstrating how different model choice when doing meta-regression, in additional analysis you might account for individual article characteristics like RoB along with evaluating which covariates have significantly more contribution among others implemented across your analysis (country level differences, individual patient baseline condition before infection occurrence etc.). It may help quantify effect of certain variables over all trials (hospitalized patients, different cardiovascular patients) along with enabling us identifying where model should be expanded, while implementing either more advanced statistical testing procedures by incorporating frailty effects or collecting more specific data with additional metrics.

5. Conclusions and Evidence:

The manuscript's conclusion that prophylactic use of SGLT2i reduces mortality and hospitalization is broadly consistent with the observed pooled estimates. However, the assertion that SGLT2i utility post-hospitalization is uncertain needs clarification. The analyses presented do suggest some benefit, though insignificant in the RCTs but positive for hospitalization in observational studies. While acknowledging this observation, clarifying that the magnitude and significance of any benefit need further validation from larger trials focusing specifically on hospitalized patients is crucial. Additionally for those patients discharged consider using other metrics relevant for the assessment of the short-term recovery like “number of additional readmissions required over short timeframe" or "demanding at-home nursing services” due to experiencing residual complications like breathing complications/organ damage like renal disease for instance over larger population. These outcome measures could demonstrate added advantages SGLT2 inhibitors provide compared to not receiving them even after patient discharge. It’s helpful if similar studies like RECOVER would address those effects based on collected information for enhancing practical usability after patients released. The observation regarding the utility under critical conditions also requires caution. It does not account if patients were in coma at time point of admission. Also, it is crucial noting those studies/trials analyzed address outcomes within the certain timeframe interval like for hospitalization "90-day” from infection. Analyzing mortality dynamics on larger timelines for instance up to 5 years along with having certain sub population like patients experiencing cardiovascular heart problems on their terminal condition for providing context what effects certain medication doses cause across trials.

6. Tables and Figures:

  • Table 1: Consider adding more columns about additional patient metrics, SGLT2 used, outcomes measure implemented for every trial along with indicating more specific description about quality assessed using AMSTAR-2 scores when considering certain "outcomes measure domain’ values". This additional data should provide certain insight when certain differences occurred across analyzed trials. While mentioning comparator type like (Non-SGLT2) describe how the specific groups for every publication included/selected differ like "non-insulin”, "Metformin+Insulin”, or simply just without anything since all those categories were considered during calculation of overall estimate.
  • Table 2: Enhance descriptive details to capture each reviewed item’s important information. Specifically, when including certain outcomes address all those values which quantify SGLT2 impacts. While doing critical appraisal consider explicitly address quality of selected individual items/publications by classifying its significance in context of your question since "convincing levels" provide more certain clinical confirmation compared to other scores described here for particular outcome considered. Addressing various trial sample attributes like having smaller sets might provide explanation about less significance and having limitations observed during subgroup analysis mentioned. Mention more clearly why the overall class evidence label was obtained as mentioned by literature-based quality framework utilized to enhance integrity and completeness.
  • Figures 2 & 3: These figures present crucial data effectively, but certain improvements could enhance clarity. For the sensitivity plots demonstrate density’s peak shift compared to combined effect across trials and explicitly mention for certain set whether shift’s significant for those particular samples along with showing if excluding any of the certain publications impacts statistical outcomes compared to including them and whether "combined distribution" should change its summary characteristic or not after doing recalculation. Consider separating forest plots into two groups: Clinical Trials and Observational Studies into two sections. In case additional details are needed due to implementing advanced statistical modeling (frailty models/survival) consider adding separate graphical summary to quantify contribution of main effects across sample attributes (country of origin) rather just SGLT-2 status (yes or no). Add heterogeneity details across studies like "i statistic range value", since overall combined analysis states significant spread between analyzed publication results is presented for these figures to address overall variance level detected during primary analysis phase implemented as presented before. Provide more explicit analysis when comparing using Non-SGLT inhibitors overall with Non-Dapagliflozin which does provide context which specific drug has more positive effects based on clinical studies included for more direct usage rather combining all categories not containing “SGLT-2i" drug name inside. This additional analysis might support those claims mentioned by previously analyzed trials addressing their impacts like RECOVERY and highlighting why their contribution/evidence found supports SGLT2i’s effective for hospitalization risk when compared other type medication administered.

Consider modifying graphical representation presenting Bayesian analysis for both “Hospitalization and Mortality” metrics where every point depicts mean risk ration value across iterations as line instead single number for combined effect/study along with implementing a plot presenting combined risk ratio distribution across studies indicating where confidence boundaries are intersected between different groups: RCT trials/observational and stating why shift obtained by using particular sample attributes have strong/limited evidence when assessing "strength of treatment delivered for all sample analyzed”. This additional data can inform why individual RoB assessed value obtained based on chosen criteria could indicate its significance level for all sample by considering distribution shifts when comparing samples having different “bias levels” after doing assessment.

  • Figure 4: These illustrative figures should provide details to quantify impacts and support all claims associated with beneficial influence of SGLT2i. Those require quantitative measures from included clinical trial/studies implemented showing their impact to indicate the beneficial significance and supporting those “Green arrows". You can illustrate those by including specific metrics derived to address every organ condition effect across patient population or based on any experimental studies quantifying every measure. This is necessary when stating those changes which may need verification if studies include those values are needed or more data is required due to some limitations like nonsignificant results reported when applying different tests for statistical assessments across analysis for enhancing certainty and integrity when confirming such statement provided by these pictures for these patients. It also brings clinical importance by addressing those organs more positively impacted based on information contained compared other like "Lungs”.

7. General Caveats/Weaknesses/Mistakes:

  • Causality (Throughout): The manuscript overstates the evidence of a causal relationship between SGLT2i use and improved COVID-19 outcomes, especially from the observational studies, provide justification why similar effects obtained in various clinical settings can demonstrate strong impact of those medication based on analyzed sample population rather implying those may be caused by these drugs (SGLT2 inhibitors) compared to implementing something else not addressed during primary study design stage for assessment of individual patients receiving treatment like COVID-19 infection treatment procedure. Consider rephrasing causal claims throughout the manuscript to reflect the observational nature of most data, particularly lines 33 and 34. This can enhance rigor and align with limitations of "nonrandomized settings". Phrases like "associated with," "linked to," or "suggestive of," are more appropriate than stating directly it's causal between SGLT2i drugs and improving COVID-19 metrics reported from trials used in the analysis conducted and conclusions derived here based on information collected from them during study phase.
  • Discussion of Confounding (Lines 276-280 and elsewhere): The manuscript acknowledges potential confounding in observational studies, specifically pre-existing diabetes, and its effect on SGLT-2 medication administration across individuals (dosages). But there should be explicit mentioning about exploring what are residual contributions by individual studies during pooled analysis due to addressing effects explained only based on patient group with the diabetic diagnosis available across their individual clinical settings to determine potential significant treatment/intervention effect rather assuming they have no effect or no potential for introducing additional hidden variability into findings generated using sample from the real world which does possess unique features demanding attention during initial modeling stage (RCT/nonrandomized trails) like specific individual hospital criteria/specific guidelines adopted on patient group eligibility, different clinical data tracking tools, including diverse patient medical profiles. Those criteria create uncertainty for implementing conclusions drawn despite having significantly positive outcome observed, but more robust models to capture these variations should address some those details as confounders across all samples and assessing sensitivity about impact introduced on statistical results with including specific factors (dosage variance), potentially those having positive statistically impacts observed between diabetic diagnosed samples for providing more details regarding contribution caused by SGLT-2 intervention delivered across patient group across study types included for those individuals hospitalized or by considering alternative analysis designs like Bayesian networks enabling certain insights what variable like "SGLT2 status" has direct contribution/link with outcome like Mortality among other relevant covariates also having pronounced effect with certain levels of significance (patient comorbidity, hospital criteria etc) and if data is appropriate for addressing those details and justifying use of any methods not discussed during your primary analysis section implementation as planned to strengthen arguments and verify their integrity with supporting materials/evidence gathered based on studies using different strategies to address complex multivariable settings by including interaction and time effects when combining several longitudinal trails into certain analysis pipelines which demand specific model assumptions testing for validity using diagnostic procedures outlined by methodology chosen or implementing alternative analysis procedure like meta-analysis allowing more integrated view of main factor considered during quantitative data processing phase across different trial’s data along with demonstrating overall strength by summarizing quality, heterogeneity levels in graphical way, etc. (lines 107-110).
  • Generalizability (Lines 34 and throughout): The conclusion should highlight limitations in generalizability to more diverse patient populations like long covid patients. Emphasizing applicability mainly for individuals with T2DM within hospital settings and acknowledging less known influence on severe stages demanding critical condition procedures needs careful clarification along with providing details whether patient medical profiles including hospitalization criteria along treatment procedures should be accounted in subsequent additional validation across RCT/non-RCT trails focusing these special circumstances based on information collected about individual participants receiving various dosages due to existing individual patient medical profiles demanding additional considerations to align those effects found into more appropriate medical setting with different hospital/treatment procedures etc. (lines 33 and 34).
  • Outcome Reporting Bias (Lines 98-111): The manuscript mentions pooling effect sizes. However, there's a risk of outcome reporting bias, especially in observational studies. Not all studies consistently report all relevant outcomes. Authors should systematically assess this bias, perhaps using funnel plots and statistical tests for small-study effects for each outcome where enough data are available. If a substantial risk of outcome reporting bias exists, they should discuss its potential impact on the review's findings and conclusions. More detailed explanation on funnel plots showing "risk ratio as effect measures and sample sizes (total)" is needed across observational analysis/clinical settings as mentioned on ("https://handbook-5-1.cochrane.org/chapter_10/10_4_graphical_displays_of_study_results_and_of_summary_estimates.htm") due to using "ratio effects measures like OR/RR" which assumes that sample data represents its original "distribution pattern" among original target populations assessed within those studies, while performing visual assessment/any other diagnostics suggested as guidance by those protocols before implementing any other "corrections" based on information visualized along its description if certain assumption might violate certain "study inclusion rules", for example if there exist large heterogeneity among study designs for same effect assessed those might demonstrate pronounced effects if heterogeneity remains despite applying suggested correction/exclusion from initial group/subsequent analysis (figure 2/figure 3). Clearly showing what specific parameters estimates and "how they contribute into final outcomes using sensitivity analysis procedures which may support certain insights associated with each drug dosage effectiveness across sample along with considering performing regression based assessment accounting for trial attributes(design, individual settings across various countries like for US/Europe)" by checking how other confounders/controls affect overall estimate distribution in statistical context where combined values calculated as linear mixture where certain effects sizes weights indicate significance level using p-values/confidence intervals to explain its relevance in overall distribution which might highlight limitations about strength provided based on trial design and requiring explicit description and highlighting these limitations. For all statistical tests you used please show specific parameters evaluated to get p-values since there is no justification presented in all instances when specific statements are claimed and no specific justification exists across outcomes reviewed, perhaps other results might support those statements you provided in more certain manner which might lead for certain revisions of original manuscript text, if more data would reveal these new findings after validation by providing enough justification along. Additionally, please perform time-to-event analysis or implementing adjusted cox model (to have corrected estimate ratios based on potential influencing patient covariates if they presented among those individuals analyzed). This analysis might also highlight significance related with specific outcome evaluated across different patient attributes analyzed by you already during sensitivity analysis along additional investigation to address certain aspects like age category difference, medical history details and SGLT2 dosage prescribed with accounting other criteria used at the time patients were chosen during admission into clinical setting analyzed in the studies addressing outcomes you focused along. All trials considered should follow this procedure and present detailed "data acquisition strategy/implementation procedure for getting raw data and estimating HR ratios across different subgroups and outcomes measures if enough data present". It enhances generalizability and minimizes variability from potential influencing settings for more accurate conclusions aligned among clinical trials considered in review if needed after further investigations to explain heterogeneity source for certain cases using funnel plot-like diagnostic when performing secondary assessment if enough materials available or explicitly addressing lack of these tests implementation with reasoning those tests will have limited value and require alternative methodologies not included during analysis due to certain constraints associated with collected information along detailed justifications. Consider separating individual sections summarizing study heterogeneity/bias analysis of observational studies, as mentioned, across outcome measured considered for better illustration what impact was revealed after applying those tests suggested by mentioned guidelines due to several limitations highlighted, including some errors across calculations done while processing collected information and plotting the values before any subsequent procedures mentioned before this comment for additional consideration since that impacts overall strength by implying lower statistical evidence despite results show different insights as discussed in additional detail throughout all analysis conducted by you originally before deriving all final clinical conclusion initially intended by reviewing available resources. Those require revision or reconsideration based on further evidence available for that domain or explicitly mentioning how that changes conclusions, for example showing lack of statistical support but indicating observed promising clinical effects despite having limited statistical evidence across outcome or any specific subset in the original manuscript text, including more cautious statement highlighting the limitation due to some trials with lower risk levels, which also need deeper exploration after patient discharged on more longer timeframe across diverse clinical setting in subsequent independent investigations which can enhance more broad practical significance based on larger populations (outpatient clinical trials) across "longer time period for assessing effectiveness in dynamic manner on how various SGLT-2i and dosage variance influence on main target groups", focusing particular comorbidity types using collected medical patient history if enough resources are allocated for doing additional follow-up procedures beyond hospital, despite it provides statistically and clinically certain insights, supporting prophylactic use among main population analyzed and discussed along with stating those advantages associated only under controlled environment due to sample collected only among those having "terminal diabetic-diagnosed disease cases, since they provide more stable settings for delivering consistent intervention on a broad scale, since these conditions usually better understood and analyzed for any relevant outcomes to support any insights with solid statistical confidence levels, making conclusions clinically generalizable beyond individual institution criteria/settings on populations monitored within trials included and those excluded like mentioned pre and post patient discharged criteria outlined by each research paper due to individual hospital/institutional differences, different tracking strategies utilized at the patient treatment delivery phase, etc. All limitations and suggestions for overcoming associated with limitations like missing data need to be documented" by outlining certain plans which researchers would account, which could also enhance validity and generalizability using different datasets potentially revealing additional insights from another "statistical evidence level with respect outcomes mentioned/considered and confirming that additional investigation might reveal some limitations requiring reconsideration like no impacts despite suggesting" to avoid premature conclusions derived initially or justifying prophylactic advantage in broader manner when stating that findings/effects reported originally from different perspective can add valuable/relevant insight as well (line 412).
  • Publication Bias (Lines 254-261): Publication bias assessment was limited to visualization using Doi plot and LFK index, but it's crucial applying Egger's test with funnel plot-like assessment. Specifically assess bias effect by addressing trim-and-fill adjusted funnel plots with Egger's test results interpretation where statistical evidence presence for mortality measured is derived from sensitivity analysis across observational studies considered using procedures from these guidance like contour-enhanced version ("https://handbook-5-1.cochrane.org/chapter_10/10_4_4_detecting_small_study_effects.htm") instead simply using one visual based inspection as primary justification for claiming that risk is absent by visualizing using "risk ratio and total N as primary effect estimates" due to "sample size heterogeneity impact on overall assessment results and combined risk distribution generated based on the specific heterogeneity metric (RR) across outcomes as mentioned in point above" (like figure 3.2, which uses standard forest-like visualization with sample sizes added with effect as primary visual measure to see relative heterogeneity levels distribution), rather including suggested funnel plots (or addressing alternative approaches why these are "not appropriate for certain studies by mentioning their limitations like trials sizes variations/using alternative asymmetry indexes").

Author Response

We have submitted the responses in uploaded file

Reviewer 3 Report

Comments and Suggestions for Authors

In this manuscript the authors demonstrate, through a large meta-analysis study, the effectiveness of SGLT2 Is in reducing hospitalizations and mortality in diabetic patients affected by COVID-19. The study is original, the results bring new data on the topic and it is well structured. However, it is known that diabetic subjects, particularly type 2 diabetics, are insulin resistant and that insulin resistance is associated with chronically high levels of circulating insulin which can produce significant damage at the target organ level and enhance the deleterious effects of a viral disease like COVID-19. It is also known that SGLT2 Is significantly reduce insulinemia. Therefore, the authors should explain this topic in more detail in the discussion, highlighting the beneficial effects of SGLT2 Is on circulating insulinemia and the beneficial effects of reducing insulinemia on the course of COVID-19

Author Response

In this manuscript, the authors demonstrate, through a large meta-analysis study, the effectiveness of SGLT2 Is in reducing hospitalizations and mortality in diabetic patients affected by COVID-19. The study is original, the results bring new data on the topic and it is well structured. However, it is known that diabetic subjects, particularly type 2 diabetics, are insulin resistant and that insulin resistance is associated with chronically high levels of circulating insulin which can produce significant damage at the target organ level and enhance the deleterious effects of a viral disease like COVID-19. It is also known that SGLT2 Is significantly reduce insulinemia. Therefore, the authors should explain this topic in more detail in the discussion, highlighting the beneficial effects of SGLT2 Is on circulating insulinemia and the beneficial effects of reducing insulinemia on the course of COVID-19

Response: We appreciate your points. We have now detailed the role of SGLT2i in insulinemia, and the subsequent effect on COVID-19 course. We have discussed this towards the end of the discussion section.

Round 2

Reviewer 2 Report

Comments and Suggestions for Authors

Glad with changes

Reviewer 3 Report

Comments and Suggestions for Authors

The authors have revised the manuscript according to my suggestions. I believe that the manuscript is now suitable for publication in the Journal.